# Bright and Early: Inhibiting Human Cytomegalovirus by Targeting Major Immediate-Early Gene Expression or Protein Function

**DOI:** 10.3390/v12010110

**Published:** 2020-01-16

**Authors:** Catherine S. Adamson, Michael M. Nevels

**Affiliations:** School of Biology, Biomedical Sciences Research Complex, University of St Andrews, St Andrews KY16 9ST, Scotland, UK

**Keywords:** herpesvirus, cytomegalovirus, immediate-early, IE1, IE2, antiviral, ribozyme, RNA interference, CRISPR/Cas, small molecule

## Abstract

The human cytomegalovirus (HCMV), one of eight human herpesviruses, establishes lifelong latent infections in most people worldwide. Primary or reactivated HCMV infections cause severe disease in immunosuppressed patients and congenital defects in children. There is no vaccine for HCMV, and the currently approved antivirals come with major limitations. Most approved HCMV antivirals target late molecular processes in the viral replication cycle including DNA replication and packaging. “Bright and early” events in HCMV infection have not been exploited for systemic prevention or treatment of disease. Initiation of HCMV replication depends on transcription from the viral major immediate-early (IE) gene. Alternative transcripts produced from this gene give rise to the IE1 and IE2 families of viral proteins, which localize to the host cell nucleus. The IE1 and IE2 proteins are believed to control all subsequent early and late events in HCMV replication, including reactivation from latency, in part by antagonizing intrinsic and innate immune responses. Here we provide an update on the regulation of major IE gene expression and the functions of IE1 and IE2 proteins. We will relate this insight to experimental approaches that target IE gene expression or protein function via molecular gene silencing and editing or small chemical inhibitors.

## 1. Introduction

Human cytomegalovirus (HCMV), also known as human herpesvirus 5, is a member of the *Betaherpesvirinae,* a subfamily of the *Herpesviridae.* Infectious HCMV particles are composed of a polymorphic lipid envelope containing viral glycoproteins, a tegument layer consisting mainly of viral phosphoproteins and an icosahedral protein capsid encasing the viral genome [1,2]. The HCMV genome comprises roughly 235,000 base pairs of double-stranded DNA in a single chromosome. By harnessing cellular RNA polymerase II, the viral genome gives rise to a highly complex transcriptome encompassing both mRNAs with more than 700 translated open reading frames as well as non-coding RNAs [3,4,5,6,7,8,9]. Upon infection of permissive cells, the HCMV genome is expressed and replicated in three sequential steps referred to as immediate-early (IE), early and late. The viral major IE gene, expressed within hours of infection, and the corresponding IE proteins will be at the center of this review. Major IE proteins inhibit intrinsic and innate host cell responses and initiate transcription from viral early genes [10,11,12,13,14,15]. Early gene products regulate host cell functions to facilitate virus replication and contribute to late events including viral DNA replication and packaging. Typical early viral proteins include the DNA polymerase (pUL54), phosphotransferase (pUL97) and components of the terminase (pUL51, pUL52, pUL56, pUL77, pUL89, pUL93, pUL104), which are all targets of approved anti-HCMV drugs [16,17,18]. Finally, late genes are expressed after viral DNA replication has commenced and encode mostly structural proteins of the capsid, tegument or envelope required for the assembly and egress of progeny virions [19,20,21]. HCMV replicates in a wide variety of differentiated cell types, and targets select types of poorly differentiated cells including myeloid progenitors for latent infection with limited viral gene expression [22,23,24,25,26]. Viral reactivation from latency is brought about by cellular differentiation and/or stimulation and contributes greatly to pathogenesis in vulnerable hosts [27,28,29].

HCMV is the cause of an ongoing “silent pandemic” affecting 40% to 100% of people in populations around the world. Co-evolution over millions of years has resulted in latent or low-level productive HCMV infection that persists for the life of the host in the absence of major disease symptoms. This type of persistence is due to a fine-tuned balance between our intrinsic, innate and adaptive immune responses and manifold viral countermeasures. Developmental or acquired immune system defects disrupt the delicate balance between virus and host and can result in severe disease outcomes. HCMV infection is the most common congenital (present at birth) infection worldwide, with an estimated incidence in developed countries between 0.6% and 0.7% of all live births. This incidence results in approximately 60,000 neonates born every year with congenital HCMV infection in the United States and the European Union combined [30,31,32,33]. Since congenital HCMV infection parallels maternal seroprevalence, the estimated incidence in developing countries is even higher, between 1% and 5% of all live births [34,35]. More than 10% of congenitally infected children will suffer neurodevelopmental damage and other disorders present at birth or long-term sequelae including hearing loss. Consequently, HCMV has been recognized as a leading cause of birth defects. HCMV reactivation from latency or primary infection also remain a major source of morbidity and mortality in immunosuppressed individuals including recipients of solid organ and haematopoietic stem cell allografts, people with acquired immunodeficiency syndrome (AIDS) and other critically ill patients. For example, HCMV infections are diagnosed in roughly 50% of all allograft recipients [36,37,38]. Cytomegaloviruses are highly species-specific, but certain aspects of HCMV infection and pathogenesis are replicated in animal models including mice infected with murine cytomegalovirus (MCMV) [39,40].

HCMV is spread through various routes including sexual contact, organ and stem cell transplantation, breast milk and from mother to baby (transplacental) during pregnancy. Women can reduce HCMV transmission through practicing appropriate hygiene behaviors [41,42,43,44]. In seropositive pregnant women HCMV hyperimmunoglobulin is applied as passive immunization to improve the adaptive immune response and reduce the risk of congenital infection. However, the value of this treatment is controversial with limited data supporting improved clinical outcomes [45,46,47,48,49]. The development of active immunization for HCMV is a major public health priority, and a number of candidate vaccines have been evaluated in clinical trials as well as preclinical models. However, no effective vaccine for HCMV is currently available [50,51,52,53,54,55,56].

Since cell-mediated adaptive immunity is believed to be key in counteracting HCMV infection, adoptive transfer of virus-specific T cells holds promise for antiviral therapy [57,58,59]. In addition, a multitude of antiviral agents from a wide diversity of chemical classes are known to be active against HCMV. The exact mechanism of action is unknown for most of these antivirals, and only a small subset has been tested in clinical trials. Seven anti-HCMV drugs have received approval for various indications: Ganciclovir (GCV), Valganciclovir, Acyclovir, Foscarnet, Cidofovir, Letermovir and Fomivirsen [60,61,62,63]. Fomivirsen is an antisense oligonucleotide targeting expression of a major IE protein and will be discussed in Section 5.1. GCV, an acyclic analogue of deoxyguanosine, was the first drug approved for the prevention and treatment of HCMV disease. The prodrug Valganciclovir is an orally applicable valine ester that is rapidly metabolized to GCV. Inside cells, GCV is converted to the active triphosphate via initial phosphorylation by the HCMV phosphotransferase (pUL97) and subsequent phosphorylation steps by cellular kinases. GCV inhibits the HCMV DNA polymerase (pUL54) by competing with deoxyguanosine triphosphate for the enzyme’s active site, thus preventing nucleotide incorporation into the elongating viral DNA [60,64]. A closely related nucleoside analogue, Acyclovir, is potent against members of the *Alphaherpesvirinae* but exhibits very modest antiviral activity for HCMV [60,65]. Thus, GCV and Valganciclovir have been the first line choice for prevention and treatment of HCMV disease. However, GCV is associated with serious toxicity including neutropenia, thrombocytopenia or anaemia [66,67]. In addition, the development of GCV-resistant HCMV strains associated with prolonged exposure, severe immunosuppression, suboptimal GCV doses and high viral loads poses a serious challenge. In 90% of all cases, resistance to GCV arises from mutations in conserved regions of either pUL97 or pUL54 [67,68]. In such cases, Foscarnet or Cidofovir are the usual alternative treatments. The two drugs do not require phosphorylation by pUL97 for activation and exhibit broad spectrum antiviral activity against DNA viruses. Foscarnet, a pyrophosphate analogue, directly inhibits pUL54 by blocking the enzyme’s pyrophosphate binding site via a non-competitive mechanism. By this mechanism, Foscarnet interferes with cleavage of the pyrophosphate moiety from the nucleotide triphosphate substrate during incorporation into the nascent DNA chain. Cidofovir is an acyclic nucleoside phosphonate and an analogue of deoxycytidine monophosphate. After phosphorylation to the active diphosphate (a deoxycytidine triphosphate analogue) by cellular kinases, competitive incorporation into the elongating DNA chain by pUL54 inhibits viral genome replication. Resistance to Foscarnet and Cidofovir occurs with rates similar to GCV, and the two drugs can select for mutations conferring cross-resistance to GCV. Moreover, lack of oral bioavailability as well as serious side effects including nephrotoxicity have limited their clinical use [66,67]. Brincidofovir is a lipid ester prodrug of cidofovir with improved oral bioavailability and reduced toxicity that demonstrated promising results in clinical trials with allogeneic stem cell transplant recipients seropositive for HCMV [67,69]. Likewise, the benzimidazole L-riboside Maribavir, a highly specific inhibitor of pUL97, has been successfully tested in clinical trials with a similar group of patients [67,69]. Recently, a number of molecules have been discovered that inhibit the packaging of viral DNA into preformed capsids by the HCMV terminase complex. Letermovir is the first of this class to be approved and reduced the levels of HCMV DNA in stem cell transplant patients in the absence of myelotoxic effects [18,70]. Although successful in immunosuppressed patients, none of the anti-HCMV drugs described above have been approved for use during pregnancy because of their teratogenic or embryotoxic effects in animal studies [60,71].

Due to the medical importance of HCMV, the absence of effective ways to prevent infection and the shortcomings of existing therapeutic drugs, it is imperative to develop novel antiviral strategies involving new molecular targets and mechanisms of action. All approved drugs currently available to prevent or treat HCMV disease target viral enzymes expressed in the early phase and critical for late processes in the infection cycle. In contrast, molecular events before the onset of HCMV DNA replication have been largely neglected with respect to antiviral approaches. This review will outline our current understanding of the regulation at the HCMV major IE gene and the functional characteristics of IE proteins derived from this gene. We will further highlight past, present and future antiviral strategies aimed at IE gene expression and protein function for improved intervention with HCMV infection and disease.

## 2. Major IE Gene Expression

### 2.1. Transcriptional Control of the Major IE Gene

The outcome of HCMV infection is believed to depend largely on the level and timing of expression from the major IE gene [23,72,73]. This is the first viral gene to be transcribed following initial infection, and likely during reactivation from latency, in a process that does not require de novo viral protein synthesis [13,23,74]. Expression of the major IE gene is highly dynamic with transcription levels ranging from extremely high to negligibly low depending on the type and differentiation or activation state of the infected cell. While productive HCMV infection is linked to activated transcription, viral latency is characterized by transcriptional repression at this gene. The major IE gene is located in the unique long (UL) segment of the viral genome, close to the internal repeat elements. The organization of this gene is unusually complex, not just by viral standards, as multiple promoters and numerous transcripts including both sense and antisense RNAs have been identified in this region [3,75,76,77,78,79,80,81,82,83]. Some of these promoters appear to have a specific role during latent infection or reactivation from latency [75,79,84]. However, the combined major IE enhancer and promoter (MIEP) is considered the principal driver of IE transcription during productive HCMV infection. It contains an extremely strong enhancer which has been widely utilized in heterologous expression systems. The MIEP is bidirectional and has been roughly divided into four functional entities: a core promoter (+1 to −40 nucleotides from the transcription start site), an enhancer (−40 to −550 nucleotides), a unique region (−550 to −750 nucleotides) and a modulator (−750 to −1140 nucleotides) [85,86] (Figure 1). The modulator’s role is largely unknown, although a cell-type specific regulatory function has been suggested [87,88,89]. “Rightward” transcription from the MIEP is suppressed by the unique region which binds cellular homeobox proteins and appears to function as an insulator between the enhancer and UL127 [90,91,92,93,94,95] (Figure 1). The core promoter is sufficient, yet not required, for low-level transcription to the “leftward” direction of the major IE gene [76,96]. It contains a TATA-box as well as the *cis*-repressive sequence (crs) that serves as a binding site for IE2 dimers (see Section 3.1). The enhancer hugely augments transcription from the major IE gene, in part via a number of small *cis*-acting repeat sequences (18-bp, 19-bp and 21-bp), and is required for viral replication. It may be further divided into proximal and distal enhancer halves (−40 to −300 nucleotides and −300 to −550 nucleotides, respectively) that differ in structural makeup, yet function jointly by contributing multiple *cis*-acting elements to provide efficient MIEP activation and viral replication. Accordingly, a long list of activating cellular transcription factors have been shown or proposed to bind to the *cis*-acting elements in the enhancer, unique region and modulator. In addition, binding of several repressive cellular transcription factors to the enhancer and modulator has been reported (Figure 1). The vast number of transcription factors that may activate or repress the MIEP is thought to account for much of the highly dynamic expression observed at the major IE gene [13,23,74].

The complexity of MIEP regulation further amplifies when considering transcription in the chromatin or “epigenetic” context. The MIEP may undergo limited DNA methylation, especially in systems for transgene expression, and the major IE gene exhibits CpG dinucleotide suppression [97,98,99,100,101]. Beyond these observations, there is little evidence that MIEP activity or IE transcription are regulated by DNA methylation following HCMV or MCMV infection [102,103,104]. Nuclear HCMV genomes form nucleosomes, octamers of core histones H2A, H2B, H3 and H4 wrapped with just under 150 bp of DNA, resembling host chromatin structure [9,105]. Consequently, the chromatin of HCMV and other DNA viruses that replicate in the nucleus is subject to regulation by nucleosome occupancy, histone composition and post-translational histone modification [106,107,108]. Nucleosome occupancy on the MIEP is believed to be low during productive infection, but likely increases during establishment of latency based on findings from the mouse model and by analogy to other herpesviruses [9,105,109]. Numerous studies have shown correlations between activating or repressive histone modifications associated with the major IE gene and the levels of viral gene expression. For example, association of the MIEP with H3K4me2, H3K4me3, H3K9/14ac, H3S10ph or H4Kac has been linked to high levels of IE (or transgene) transcription and productive infection or reactivation from latency [72,110,111,112,113,114,115,116,117,118,119,120]. By contrast, the presence of H3K9me2, H3K9me3 or H3K27me3 at the MIEP generally correlates with low levels of IE transcription and either latent or the onset (pre-IE phase) of productive infection [110,111,112,115,116] (Figure 1). In agreement with these observations, histone modifying enzymes and enzyme complexes including histone acetyltransferases (e.g., KAT6A/MOZ), histone deacetylases (e.g., HDAC1, HDAC3), histone methyltransferases (e.g., EHMT2/G9A, EZH2, SETDB1, SUV39H1), histone demethylases (e.g., KDM1A/LSD1, KDM4A/JMJD2, KDM6B/JMJD3) and histone kinases (e.g., MSK family) have all been implicated in regulating transcription from the MIEP [72,119,120,121,122,123,124,125,126,127,128,129] (Figure 1). The histone modifying proteins are typically recruited by transcription factors bound to the MIEP including cAMP responsive element binding protein 1 (CREB1), ETS2 repressor factor (ERF) and Ying Yang 1 transcription factor (YY1). In turn, chromatin modifications lead to the recruitment of further activators or repressors that may affect IE expression making for a complex hierarchy of transcriptional regulation [107,130,131].

Histone deacetylases, histone demethylases and other proteins conferring repressive histone modifications to HCMV chromatin may be considered components of the intrinsic cellular immune system also known as restriction factors [132]. Many of the best known restriction factors for HCMV reside in nuclear organelles referred to as nuclear domain 10 or promyelocytic leukaemia (PML) bodies [133,134,135]. While PML bodies may confer transcriptional repression as a whole, constituents of these organelles including alpha thalassemia/mental retardation syndrome X-linked protein (ATRX), death domain-associated protein (DAXX), PML protein and SP100 nuclear antigen have been shown or proposed to act as repressors of major IE gene expression in part via chromatin-based mechanisms [136,137,138]. More recently, cellular proteins that mediate foreign or damaged DNA sensing and signalling, including cyclic guanosine monophosphate-adenosine monophosphate (cGAMP) synthase, interferon (IFN) gamma-inducible protein 16 (IFI16) and stimulator of IFN genes (STING), have been identified as restriction factors of HCMV and other DNA viruses [139,140,141,142]. These proteins are known or predicted to restrict IE transcription, at least indirectly, although IFI16 may activate rather than repress the MIEP [143,144].

Expression of the major IE gene also varies with the activity of cellular signalling pathways that connect the extra- and intracellular environment to the nucleosomes and transcription factors associated with the HCMV genome including the MIEP. The virus has been shown to activate, rewire or inhibit numerous of these signalling pathways. HCMV infection triggers both pathways considered to be proviral as well as pathways linked to innate immune responses resulting in the production of proinflammatory and antiviral cytokines. In fact, many signalling pathways appear to exhibit both pro- and antiviral potential, and the net effect on the virus depends on various factors including cell type and stage of infection. Binding of HCMV to receptor proteins on the cell surface initiates the first wave of signalling. The virus engages various cellular entry receptors, several of which activate similar pathways relevant to the IE phase of infection [145,146,147,148,149]. In particular, epidermal growth factor receptor (EGFR), platelet-derived growth factor receptor alpha and integrins independently trigger the phosphatidylinositol 3-phosphate and protein kinase B (PI3K/AKT) pathway [150,151,152]. The PI3K/AKT pathway is central to many cellular properties including motility, proliferation and survival [153,154,155]. Transient induction of this pathway triggered by receptor signalling appears to be followed by more sustained activation involving the viral major IE proteins [156,157,158,159]. Initial PI3K/AKT activation is required for efficient viral entry as well as optimal replication in fibroblasts and establishment of latency in monocytes [156,158,160,161,162,163]. However, at later times during infection inhibition of EGFR or PI3K seems to favour viral replication and reactivation from latency suggesting a negative regulatory role at this point [164,165,166,167]. Besides PI3K/AKT signalling, various other kinase pathways are known to be activated very early during HCMV infection. These pathways include mitogen-activated kinase (MAPK) signalling both via extracellular signal-regulated kinase (ERK) 1 and 2 including RAF1 (MAPKKK upstream of ERK) as well as via p38 MAPK [168,169,170,171,172]. Other kinases thought to be involved in the IE phase of HCMV infection include adenosine monophosphate-activated protein kinase (AMPK) [173], hematopoietic cell kinase (a src family kinase) [174], cyclin-dependent kinases (CDKs) [175], protein kinase A [176] and mitogen and stress activated kinase (MSK) [128]. The activation of kinase signalling pathways in the initial infection phase comes with multiple, mostly beneficial consequences for the virus including major IE gene activation. For example, ERK mediates induction of major IE gene expression via binding of CREB to the MIEP and recruitment of MSK. In turn, MSK-mediated histone H3 phosphorylation promotes histone demethylation and the subsequent exit of HCMV from latency [128]. One of the most crucial transcription factors linked to the PI3K/AKT, MAPK and other signalling pathways relevant to HCMV infection is nuclear factor kappa B (NF-κΒ). Canonical NF-κΒ activation requires degradation of inhibitor of NF-κΒ (IκΒ), which depends on phosphorylation by a three-subunit IκΒ kinase (IKK). IKK-mediated phosphorylation of IκΒ is triggered as early as five minutes after exposure of cells to HCMV particles resulting in activation of preformed NF-κΒ [177,178]. This first phase of the NF-κΒ response to HCMV infection may facilitate IE expression via binding sites in the proximal enhancer of the MIEP (Figure 1). However, the requirement of NF-κΒ for efficient IE expression varies widely with cell type, virus strain and other conditions of infection [179,180,181,182]. A second phase of NF-κΒ activation due to initiation of NF-κΒ transcription allows for continued expression throughout infection. While NF-κΒ activation benefits HCMV replication, at least under certain conditions, it also comes with adverse effects for the virus. NF-κΒ, along with IFN regulator factor 3 (IRF3), binds to promoters and stimulates transcription of numerous cytokine and chemokine genes. Some of these genes encode antiviral proteins including type I IFNs. HCMV gene products including tegument proteins (e.g., pUL35, pUL82/pp71, pUL83/pp65) and IE proteins as well as non-coding RNAs target the IFN response and other signalling pathways, adding an additional layer of complexity. Targeting of host cell signalling by HCMV will be discussed below in the context of IE proteins (see Section 3.4), but is otherwise beyond the scope of this review. For a comprehensive and detailed account of this topic, the reader is referred to several other recent reviews [183,184,185].

### 2.2. Post-Transcriptional and Translational Control of the Major IE Gene

The primary transcript derived from the MIEP is subject to extensive regulation at the post-transcriptional and translational level. It undergoes alternative splicing and polyadenylation to generate multiple mRNA species assigned to either the IE1 or IE2 family [12,13,74]. This differential post-transcriptional regulation is believed to involve the cellular 65-kDa U2-associated factor and ubiquitin-dependent segregase valosin containing protein p97 [186,187]. RNA sequencing showed increased IE1 and decreased IE2 splicing following p97 knockdown [187]. The processed IE1 and IE2 mRNAs accumulate with different kinetics and share the first three exons [186,187,188]. However, IE1 mRNAs contain exon 4 while IE2 mRNAs contain exon 5 sequences.

Translation of the IE1 and IE2 mRNAs is subject to control by viral non-coding RNAs [189,190,191]. For example, the HCMV long non-coding RNA 4.9 has been reported to bind to the MIEP and recruit repressor complex PRC2 to this region [120]. In addition, HCMV miRNA miR-UL112-1 was shown to target the IE1 mRNA and to reduce the corresponding protein levels by translational inhibition [192,193,194]. Likewise, HCMV miR-UL25-1 and miR-UL25-2 appear to be linked to reduced IE1 protein levels, although most likely indirectly via cellular targets [195].

The major IE mRNAs ultimately give rise to the IE1 (UL123) and IE2 (UL122) families of proteins with several members each. The largest, most abundant and by far best studied family members are the 72-kDa (491 amino acids) IE1 protein, also known as IE72, and the 86-kDa (579 amino acids) IE2 protein, also known as IE86. The two proteins share 85 amino acids encoded by exon 3 at their amino termini but are otherwise unrelated. For simplicity, they are referred to as IE1 and IE2 in this review.

### 2.3. Post-Translational Control of the Major IE Proteins

IE1 and IE2 are both believed to exist as dimers, while IE2 may also form higher order oligomers [196,197,198,199,200]. Both IE proteins can undergo at least two types of post-translational modification, phosphorylation at serine or threonine residues [201,202] and conjugation to small ubiquitin-like modifiers (SUMOylation) at lysine residues [203,204,205,206]. Various positive or negative regulatory effects on IE protein function and HCMV replication have been ascribed to these modifications [205,207,208,209,210,211,212,213,214]. While IE1 is a metabolically highly stable protein with an estimated intracellular half-life between 21 and >30 h [215,216], IE2 exhibits a much shorter half-life of approximately 2.5 h in cells [197,215]. Alongside post-transcriptional mechanisms (see Section 2.2), the differences in metabolic stability contribute to the much higher steady-state levels of IE1 compared to IE2 observed during productive HCMV infection. Nuclear localization signals in IE1 and IE2 target the proteins to the cell nucleus, where they are found in various compartments including PML bodies, chromatin and the nucleoplasm [11,13,14].

### 2.4. Summary

Highly complex interactions between a multitude of cellular and viral components at the level of DNA, chromatin and upstream signalling pathways determine the initiation and magnitude of transcription from the HCMV MIEP. The highly dynamic transcription from the major IE gene is complemented by post-transcriptional processing and translational regulation, ultimately controlling the synthesis of the IE1 and IE2 families of predominantly nuclear proteins. The major IE proteins are subject to post-translational regulation and are thought to activate the viral replicative cycle during both initial infection and reactivating from latency. It is therefore believed that the eventual outcome of HCMV infection depends on the level and timing of IE1 and IE2 expression.

## 3. Major IE Protein Function

### 3.1. Role in Activation and Repression of Transcription

IE1 and IE2 were initially identified as activators of transcription in reporter assays using transiently transfected plasmids [11,13,14]. In these assays, the IE proteins were shown to activate the HCMV MIEP (positive auto-regulation) and various viral early gene promoters. In addition, IE2 turned out to be a repressor of the MIEP (negative auto-regulation). In fact, IE2 sequence-specifically binds to the crs in the core promoter (Figure 1) to block RNA polymerase II occupancy at the transcription start site. Furthermore, several heterologous viral promoters as well as cellular promoters proved to be responsive to activation by the IE proteins. The impact of IE1 and IE2 on transcription from a broad variety of promoters in transient transfection assays earned them the title “promiscuous transactivators”. IE2 usually appeared as the stronger activator compared to IE1 and, depending on the reporter construct, the two proteins often acted in an additive or synergistic manner. Activation by IE1 and IE2 was mapped to both upstream elements as well as core promoter regions including the TATA-box. Accordingly, numerous specific and basic transcription factors or transcription factor complexes were reported to interact with IE1 (e.g., CEBP, E2F1-5, SP1) and IE2 (e.g., AP1, CREB1, EGR1, SP1, TAF4, TBP, TFIIB, TFIID).

Many key findings from transient transfection assays about the impact of IE1 and IE2 on HCMV transcription were later corroborated by studies involving mutant viruses and global transcriptome analyses. These findings confirmed positive autoregulation at the MIEP by IE1 [217], crs-dependent repression of the MIEP by IE2 and activation of viral early genes by IE1 and IE2 [9,218,219,220,221]. In contrast, “promiscuous transactivation” by IE1 and IE2 was not replicated in transcriptome analyses of endogenous human genes. Instead of showing broad activation of gene expression from the human genome, the differential transcript profiles from cells individually expressing IE1 or IE2 were rather distinct with little or no overlap to the genes activated by the IE proteins in assays with transfected reporter plasmids. Following expression under conditions closely mimicking the situation during productive infection, IE1 turned out to be as significant a repressor as it is an activator of host gene expression in growth-arrested human fibroblasts [222,223]. Cells induced to express IE1 exhibited global repression of interleukin 6 (IL6)- and oncostatin M-responsive signal transducer and activator of transcription (STAT) 3 target genes. This repression was followed by STAT1-dependent activation of type II IFN-stimulated genes (ISGs), normally induced by IFN-γ, many of which encode immune-stimulatory proteins including proinflammatory chemokines [222,223,224]. Moreover, in the presence of IFN-α or IFN-β, IE1 was found to inhibit the induction of type I ISGs by the trimeric complex of STAT1, STAT2 and IRF9 known as ISG factor 3 (ISGF3) [225,226]. The effects IE1 exerts on the human transcriptome are thought to result largely from direct physical interactions with STAT2 and STAT3 (see Section 3.4). While transcriptional regulation by IE1 appears to be dominated by pathways depending on proteins of the STAT family, IE2 has been shown to inhibit the induction of IFN and other antiviral cytokine genes via a mechanism involving NF-kB and STING (see Section 3.4). However, the transcriptome profile for IE2 in cycling human fibroblasts was dominated by genes regulating the cell cycle and DNA replication many of which are E2F-responsive [227]. This finding likely reflects the impact of IE2 on the cell cycle. IE2 promotes cell cycle progression from G0/G1 to G1/S and arrests cells at the G1/S interface, inhibiting cellular DNA synthesis, or at the G2/M interface [13,14,228]. Many human genes activated or repressed by isolated expression of IE1 or IE2 were also shown to be differentially regulated during productive HCMV infection.

### 3.2. Role in Chromatin-Based Epigenetic Regulation

IE2 is known to bind sequence-specifically to DNA, but there is no convincing evidence for direct DNA binding by IE1 [11,108,131]. However, IE1 associates with chromatin via core histones. IE1 exhibits two physically separable histone interacting regions with differential binding specificities for H2A-H2B dimers and H3-H4 dimers or tetramers. The H2A-H2B binding region was mapped to an evolutionarily conserved nucleosome binding motif (amino acids 479–488) within the chromatin tethering domain (CTD) at the C-terminus [229,230]. This motif docks with the acidic patch formed by H2A-H2B on the nucleosome surface [229,230]. The consequences of the IE1-nucleosome interaction have not been fully elucidated, but they appear to include alterations to higher order chromatin structure [230]. Histone binding by IE1 might also be linked to the overall low nucleosome levels and temporal reorganization of nucleosomes across viral genomes observed during productive HCMV infection [9,105]. To our knowledge, transcriptional regulation via the IE1 CTD has not been experimentally addressed. Instead, it has been reported that IE1x4, a small variant form of IE1 expressed from a promoter internal to major IE exon 4, facilitates viral genome maintenance and replication during HCMV latency via a CTD-dependent mechanism. Despite a lack of experimental evidence, the mechanism is predicted to involve HCMV episome tethering to host mitotic chromosomes via nucleosome binding by IE1x4 resulting in nuclear retention and partitioning of viral genomes across latently infected dividing cells [75]. Although IE2 appears to have relatively little affinity for histones, both IE proteins have been shown to be present in complexes with nucleosome modifying host cell proteins. For example, IE1 binds to histone deacetylase (HDAC) 3 [124], while IE2 binds to HDAC1-3 [124,126,231], lysine acetyltransferases CREB binding protein (CBP, KAT3A), p300 (KAT3B) and p300/CBP-associated factor (KAT2B) [232,233], and lysine methyltransferases G9A (euchromatic histone lysine methyltransferase 2, EHMT2) and suppressor of variegation 3-9 homolog 1 (SUV39H1) [126]. Accordingly, transcriptional activation of viral IE and early genes by IE1 correlates with histone acetylation, while transcriptional repression of the MIEP by IE2 involves histone deacetylation and methylation [130,131,234].

### 3.3. Role in Inhibition of Intrinsic Immunity

Intrinsic or cell-autonomous immunity is considered the first intracellular line of defence against viral attack. Intrinsic immunity confers (partial) resistance to viruses via constitutively produced cellular inhibitors of viral replication known as restriction factors [137,235,236].

Consistent with its presence during the very early stages of HCMV infection, IE1, along with several viral tegument proteins, has been recognized as a viral antagonist of cellular intrinsic immunity [10,12,137]. Specifically, IE1 has been shown to target three restriction factors based in nuclear organelles known as PML bodies (see Section 2.1): PML (tripartite motif 19) proteins, SP100A and DAXX. Although various activities have been linked to these restriction factors, they all seem to mediate transcriptional repression of HCMV gene expression in part via chromatin-based mechanisms [134,137,237]. Both IE1 orthologues of animal cytomegaloviruses as well as HCMV IE1 were shown to associate with DAXX [238,239,240]. The sites of interaction in the two proteins have not been mapped, and it remains unclear whether binding is direct. However, transcriptional activation of the HCMV latent undefined nuclear antigen (LUNA) gene depends on relief from DAXX-mediated repression conferred by IE1 [238]. Most of the proteins IE1 targets remain metabolically stable, but a subset appears to be subject to proteolytic degradation [241,242,243]. IE1 was shown to interact physically with the N-terminal domain of SP100A and to target the restriction factor for degradation via the proteasome. This finding explains the loss of SP100 observed in the late phase of productive HCMV infection [241,242,244]. The relevance of IE1-mediated SP100 degradation for HCMV replication remains to be determined. Finally, it has been established that IE1 binds to PML proteins via its central core domain (amino acids 25–378) predicted to exhibit an all α-helical, femur-shaped fold [196,245]. This interaction appears to interfere with PML oligomerization and de novo poly-SUMOylation [203,246,247,248]. SUMOylated PML isoforms are the central organizers of PML bodies, and inhibition of SUMOylation by IE1 correlates with organelle disruption resulting in diffuse nuclear distribution of the associated restriction factors [203,249,250,251]. The loss of PML body integrity adds an additional layer to inhibition of intrinsic immunity by IE1 that extends beyond the mere targeting of individual restriction factors. Despite limited experimental evidence, PML targeting and disruption of PML bodies are considered to be key to IE1 function and HCMV replication, especially at low multiplicity of infection [216,252].

Preceding disruption by IE1, IE2 co-localizes with PML bodies, most likely as a consequence of binding to the viral genome which also associates with these organelles [251,253,254]. However, IE2 is not currently considered an antagonist of PML bodies. Having said that, both IE1 and IE2 target histone modifying enzymes (see Section 3.2), some of which are bona fide restriction factors, and more cellular mediators of intrinsic antiviral immunity targeted by the IE proteins will likely emerge in the future.

Finally, both IE1 and IE2 inhibit apoptotic cell death, which may be considered part of the intrinsic antiviral defence system [157,233,255,256,257,258]. It appears that each IE protein can block extrinsic apoptosis pathways via activation of PI3K/AKT signalling, although additional mechanisms likely contribute including complex formation between IE2 and p53 [259,260,261]. Despite the fact that the antiapoptotic potential of the two IE proteins has been clearly established in several overexpression settings, its true relevance to HCMV infection remains to be determined.

### 3.4. Role in Inhibition of Innate Immunity

Post-attachment events associated with HCMV entry and the recognition of virion components by pattern recognition receptors including foreign DNA sensors trigger the induction of cytokine and chemokine genes [262,263,264]. Many of these cytokines and chemokines are important components of our innate immune system, especially type I, II and III IFNs. The synthesis and secretion of these IFNs activates signalling pathways that involve the phosphorylation of STAT family members including STAT1 and STAT2. Activated STAT proteins form homodimers or heteromeric complexes that subsequently bind to and stimulate transcription from promoters of ISGs many of which encode proteins that interfere with viral replication at various points.

IE1 confers increased type I IFN resistance to HCMV [225]. This phenotype was largely attributed to nuclear complex formation between IE1 and STAT2 depending on amino acids 410–420 in the presumably disordered “acidic domain” of the viral protein downstream from the central core domain and upstream of the CTD [209,222,225,226]. The IE1-STAT2 interaction causes reduced sequence-specific DNA binding by ISGF3 and diminished activation of type I ISGs (e.g., CXCL10, IFIT2, ISG15, MX1) [209,225,226,265,266]. The C-terminal part of IE1 has also been reported to disrupt type II ISG activation by STAT1 homodimers, although IE1 is not believed to bind to STAT1 directly (only indirectly via STAT heterodimers) [222,225,226,267]. The ability of IE1 to inhibit type I ISG induction via STAT2 interaction facilitates HCMV replication and appears to be conserved among IE1 homologs of other betaherpesviruses [209,226,268]. Besides STAT2 interaction, complex formation between PML and IE1 (see Section 3.3) may also contribute to the inhibition of ISG induction during HCMV infection [216,269].

IE2 is not known to interact with STAT family members, but this protein limits HCMV-induced expression of antiviral cytokine and proinflammatory chemokine genes (e.g., IFNB1, CCL3, CCL5, CCL8, CXCL8, CXCL9) [270,271]. A very recent report has also shown that IE2 targets interleukin 1 beta (IL1B) at both the transcript and protein level [272]. The underlying mechanism appears to involve inhibition of virus- or tumor necrosis factor alpha-induced binding of NF-κB to the IFN-β promoter, resulting in attenuated target gene expression [273]. Another recent report has demonstrated that IE2 inhibits IFN-β promoter activation induced by STING, a critical sensor of intracellular DNA and adaptor for type I IFN signalling. IE2 facilitated the proteasome-dependent degradation of STING and inhibited cGAMP-mediated induction of IFNB1 and CXCL10 [274]. Taken together, these studies suggest that IE2 targets STING (and likely other proteins) post-translationally resulting in inhibition of NF-κB-dependent induction of cytokine and chemokine genes relevant to the innate immune response to HCMV infection.

### 3.5. Role in Inflammation and Adaptive Immunity

HCMV reactivation and replication are typically linked to a strong inflammatory host response that involves numerous cytokines and chemokines, which often contributes to pathogenesis [27,28,275]. Despite their roles as intrinsic and innate immune antagonists (see Section 3.3 and Section 3.4), the major IE proteins may also facilitate inflammation, most obviously by driving HCMV replication. However, IE1 and IE2 may promote inflammation even in the absence of viral replication [223,276,277,278,279,280,281]. Consistent with this idea, the IE1-specific host cell transcriptome is largely characterized by downregulation of genes responsive to IL6-type cytokines and upregulation of ISGs normally induced by IFN-γ (see Section 3.1) [222,223]. IE1-dependent gene activation proved to be independent of IFN-γ and other IFNs, yet required phosphorylated STAT1. Accordingly, IE1 induced phosphorylation, nuclear accumulation and binding of STAT1 to type II ISG promoters. Moreover, the repression of STAT3- and the activation of STAT1-responsive genes by IE1 turned out to be coupled. By targeting STAT3, IE1 rewires upstream STAT3 to downstream STAT1 signalling. Consequently, genes normally induced by IL6 are repressed while genes normally induced by IFN-γ become responsive to IL6 in the presence of IE1. Thus, IE1 merges two central cellular signalling pathways diverting cytokine responses relevant to inflammation and (neuro)pathogenesis [222,282].

Adaptive antibody as well T cell responses are thought to be important for long-term control of HCMV. Studies on T cell immunity in HCMV have traditionally focused on pUL83/pp65 and IE1. However, it has become clear that both IE1 and IE2 are highly immunogenic CD4+ and CD8+ T cell antigens adding to their complex roles in the immune response to HCMV infection [283,284,285]. Based on the stimulatory effect IE1 exerts on cellular adaptive immunity, the protein has been utilized in the development of both diagnostic assays as well as vaccine candidates [286,287,288].

### 3.6. Role in Viral Replication, Latency and Reactivation

Various highly differentiated cell types including primary human fibroblasts are susceptible to HCMV infection and permissive for viral replication. The importance of IE1 in the viral productive cycle was first highlighted by studying laboratory-adapted high passage HCMV strains (Towne/Toledo and Towne) from which major IE exon 4 had been specifically deleted. Mutant virus replication in fibroblasts was (almost) normal at high but profoundly impaired at low multiplicity of infection [217,219]. The IE1-specific phenotype was eventually attributed to a broad reduction in viral early gene expression and a failure to form replication compartments [218,219]. Nonetheless, more recent studies of IE1-deficient viruses in the background of both high (AD169) and particularly low passage HCMV strains (TB40E) demonstrated substantial attenuation even at high input multiplicity [9,216]. Thus, IE1 is important for efficient HCMV replication in cellulo, albeit not essential. In contrast, IE2 is considered to be indispensable for viral replication at any multiplicity of infection in cultured fibroblasts [220,221,289,290].

While robust expression of the major IE gene is crucial for productive HCMV infection, the absence or low levels of IE proteins are linked to the establishment of latent infection. HCMV establishes latency in a subset of typically poorly differentiated susceptible cells including cells of the myeloid lineage. The MIEP is largely repressed in these cell types, although low levels of IE1 (and even IE2) may still be produced. A study led by the late Greg Pari proposed that a variant form of IE1 referred to as IE1x4 rather than the full-length protein is expressed in latently HCMV-infected haematopoietic progenitor cells [75]. Their results suggest that IE1x4 is required for latent viral genome replication and maintenance involving interactions with the cellular transcription factor SP1 and topoisomerase IIβ. This report is in line with the idea that the IE1 CTD binds to mitotic chromosomes via the acidic patch formed by histones H2A-H2B on the nucleosome core particle (see Section 3.2). The presence and function of IE1x4 during HCMV latency await independent confirmation.

Although it is generally assumed that IE1 and IE2 are required for HCMV reactivation from latency, there is little experimental evidence to confirm this notion. In a promonocytic cell-line, ectopic expression of IE1 and IE2 was sufficient for induction of viral early gene expression but not for production of infectious virus [129]. Studies in the mouse and rat models concluded that the IE1 orthologs are not even required for viral reactivation from latency [291,292,293]. Thus, while IE2 is almost certainly necessary for HCMV reactivation (being essential for viral replication) the importance of IE1 in this process remains ambiguous.

### 3.7. Summary

IE1 and IE2 are nuclear localized HCMV proteins expressed at the beginning of infection. They autoregulate the MIEP, activate viral early genes and modulate expression of cellular genes many of which are involved in the cytokine and chemokine response to infection. Regulation of viral gene expression by the IE proteins appears to result in part from chromatin-based mechanisms including histone modification, and at least IE2 shares properties with factors that actively control transcription. In addition, IE1 and IE2 regulate transcription more passively by targeting signalling effectors upstream of the genome such as STAT2/3 and STING, respectively. Both IE proteins are powerful antagonists of intrinsic and innate immunity predicted to be individually essential for HCMV replication in vivo. That said, IE1 and IE2 may contribute to HCMV pathogenesis even in the absence of viral replication.

## 4. Case for Antiviral Targeting of Major IE Gene Expression or Protein Function

Antiviral strategies for HCMV have long relied on a single molecular target, the viral DNA polymerase. Even more recently approved antivirals and candidate drugs under development are directed at viral targets involved in late molecular processes of HCMV replication including DNA packaging. At this late stage, infection is fully established and adverse immune-related effects including inflammation have been triggered. In fact, immunopathogenic rather than cytopathogenic origins have been proposed for some HCMV disease including pneumonitis in allogeneic transplant recipients [275,276,277]. Similarly, in mouse models of pneumonitis MCMV replication was not sufficient to cause disease [276,277,278,279]. Conversely, MCMV caused pneumonitis in the absence of viral replication [280]. Likewise, HCMV retinitis in AIDS patients was proposed to be partly due to immunopathogenesis triggered by IE gene expression, as disease progressed in the absence of replicating virus [281,294]. Along these lines, the IE1 protein was shown to induce pro-inflammatory gene expression and chemokine secretion [222,223]. The chemokines produced upon IE1 expression included C-X-C motif chemokine receptor 3 (CXCR3) ligands CXCL9, CXCL10 and CXCL11, which have been implicated in a large variety of inflammatory and other immune-related disorders including transplant dysfunction or rejection [278,279]. This evidence links IE gene expression to HCMV pathogenesis.

We consider the major IE gene and proteins promising alternative or complementing targets for anti-HCMV strategies for various reasons. Targeting the expression or function of IE1/2 would interfere with infection at a “bright and early” stage before all other currently approved systemic drugs. MIEP- or IE1/2-targeted drugs are predicted to prevent or dampen inflammation even before viral replication commences. Since IE1/2 are also powerful antagonists of intrinsic immunity and the IFN response, compounds targeting their expression or function are expected to confer susceptibility to these host responses providing a novel mechanism of action. In addition, the IE1/2-targeted drugs exhibit potential for “epigenetic therapy” as both viral proteins exert their functions in part via histone modifications, again providing a novel mechanism of action. These drugs are expected to interfere not only with an ongoing productive infection but also with early stages of reactivation, since both the MIEP and IE1/2 function are likely required for this process. Conceivably, even viral persistence may be inhibited based on the observation that IE1x4 mediates viral genome replication and maintenance during latency. Finally, IE1/2-directed drugs are not expected to confer cross-resistance to or interfere with the activity of existing compounds approved for HCMV monotherapy. They may therefore be combined with these drugs for combination therapies with improved efficacy.

## 5. Inhibition of Major IE Gene Expression by Gene Silencing or Editing

### 5.1. IE Gene Silencing

Silencing IE gene expression is expected to exert significant pleiotropic antiviral effects due to the multi-functional roles played by IE gene products in HCMV replication, latency and pathogenicity (see Section 2 and Section 3). Molecular approaches can efficiently target IE gene expression (Figure 2), and initial feasibility of this approach has been demonstrated via the antisense oligonucleotide Fomivirsen (also known as ISIS 2922 or Vitravene). Fomivirsen is a 21-base synthetic oligonucleotide complementary to IE2 mRNA sequence with phosphorothioate linkages to enhance nuclease resistance. It exhibits potent HCMV antiviral activity with EC_50_ values in the sub-micromolar range [295,296]. Fomivirsen’s mechanism of action is primarily thought to block IE2 gene expression by sequence-dependent hybridization to its target mRNA that results in reduced IE2 protein levels due to mRNA degradation via RNaseH recognition of the DNA:RNA hybrid complex [295,297]. This is not, however, the sole mechanism of action, as other sequence-dependent and sequence-independent effects have been reported to contribute to its antiviral activity [295,297,298]. Fomivirsen, developed by Isis Pharmaceuticals in collaboration with Novartis Opthalmics, was in 1998 the first oligonucleotide-based therapy to be approved for clinical use by the FDA [299]. It was approved for treatment of HCMV-induced retinitis in HIV/AIDS patients via local intravitreous injection, and its clinical effectiveness was demonstrated in small-scale clinical trials [300,301,302]. Fomivirsen is no longer marketed, due to a significant decline in HCMV-induced retinitis cases in HIV/AIDS patients following the successful implementation of antiretroviral therapy and the availability of alternative treatments [299]. Despite its discontinuation for commercial reasons, Fomivirsen’s development has provided convincing proof-of-concept evidence that inhibition of IE gene expression can be an effective HCMV antiviral therapeutic strategy.

An alternative approach to targeting IE mRNA and hence IE gene silencing, is to utilize gene-targeting ribozymes, which are catalytically active RNA molecules that specifically cleave target mRNA sequences. M1GS ribozyme technology has been used to target both IE1 and IE2 by utilizing the shared mRNA region of these genes [303,304,305,306]. Target IE1/2 mRNA sequences have been selected by determining accessibility for M1GS binding via dimethyl sulfate mapping [303,304,305,306]. M1GS is partially derived from the M1 RNA catalytic subunit of the *E.coli* RNase P ribozyme, which mediates tRNA maturation [307,308]. M1 RNA can be converted into an M1SG sequence-specific ribozyme by covalently linking it to an external guide sequence (EGS) that contains nucleotides complementary to the target mRNA sequence [307,308]. The tertiary structure generated upon hybridization between the mRNA substrate and the EGS is required for recognition and cleavage by the ribozyme active site [307,308]. The initial IE1/2-targeted study used a wild-type M1 sequence and IE1/2 exon 3 as the cleavage site. This IE1/2-targeted ribozyme reduced IE1/2 gene expression by 75–80% and inhibited HCMV replication 150-fold in cell culture [303]. A protein engineering and selection strategy has subsequently been employed to identify various highly active M1SG variants containing mutations in M1 that enhance substrate binding and cleavage rates [304,305,306,309]. The most potent variant reported to-date, F-R228-IE, reduced IE1/2 expression by 98%–99% and inhibited HCMV replication 50,000-fold in cell culture [306]. F-R228-IE uses nucleotide position 43 downstream from the IE1/2 initiation codon as the designated cleavage site and contains three M1 RNA point mutations (G59A, C123U, C326U). However, the mechanism by which these mutations enhance cleavage is currently unknown [306]. Whilst M1SG technology has potential for HCMV therapeutic applications, to the best of our knowledge it has not yet been clinically tested.

RNA interference (RNAi) offers an alternative approach to targeting IE gene expression. RNAi is a cellular gene-silencing pathway that results in sequence-specific degradation of the target mRNA via complementary short-interfering (siRNA) molecules. Various siRNA or short-hairpin RNA (shRNA, processed into siRNA) molecules targeting IE1/2 mRNA have been shown to cause significant inhibitory effects on HCMV replication in cell culture. These antiviral effects correlated with reductions in IE1/2 mRNA and protein levels [310,311,312,313]. In addition, IE1/2 siRNA treatment offset some consequences of HCMV infection for the host cell, by retaining PML body integrity and preventing DNA damage response signalling [310]. Treatment of cells with IE-targeted siRNA after HCMV infection resulted in a modest antiviral effect; this is a valuable observation as therapeutic treatment of patients after establishment of HCMV infection would be an important clinical application [310]. Although RNAi technology has potential for anti-HCMV applications, this technology along with antisense oligonucleotides and gene-targeting ribozymes, may be superseded by the recent development of genome-editing techniques. 

### 5.2. IE Gene Editing

Instead of targeting IE gene expression at the mRNA level, genome-editing technology could be used to directly target the HCMV DNA genome to disrupt the UL122/123 gene responsible for major IE transcription. At the time of writing, one study has reported UL122/123 gene-editing, using the Clustered Regularly Interspaced Short Palindromic Repeats (CRISPR)/CRISPR-associated protein 9 (Cas9) system [314]. CRISPR/Cas9 is a new powerful technology that targets specific DNA sequences in eukaryotic cells for cleavage, via double-stranded DNA breaks, using a Cas9 endonuclease and a guide RNA (gRNA) that determines target specificity. Double-stranded DNA breaks are repaired by host mechanisms, such as non-homologous end joining, which are error prone and can introduce small insertion/deletion mutations at the targeted location, or larger deletions if multiple breaks are introduced. A multiplex strategy using three gRNAs targeting UL122/123 successfully excised the UL122/123 gene in 90% of all viral genomes in an HCMV-infected cell population and resulted in a significant decrease in IE protein production and 90% reduction in HCMV replication [314]. Multiplex approaches have been developed to overcome acquisition of resistance mutations in the target sequences. This study provides proof-of-concept that a multiplex anti-UL122/123 CRISPR/Cas9 system can efficiently target the HCMV genome. This system may be useful for targeting HCMV in latently infected cells, where viral gene expression is low or absent and thus mRNA is not available for other molecular approaches discussed in this section and viral proteins (e.g., DNA polymerase) are not available as a target for conventional antiviral drugs. 

### 5.3. Summary

Molecular techniques offer a promising future for development of new anti-HCMV approaches. However, a number of drawbacks must be addressed including reduction of toxicity as well as off-target and immunostimulatory effects combined with improvements in the mode, stability and efficiency of delivery vehicles and methods [315,316].

## 6. Inhibition of Major IE Gene Expression by Small Molecule Chemical Inhibitors

### 6.1. Introduction

Inhibition of major IE gene expression can be achieved by identification of small molecules that directly or indirectly inhibit activation of the MIEP and thus prevent IE gene transcription and translation (Figure 2). The complexity of MIEP regulation and its dependence on host cell signalling pathways and transcription machinery (see Section 2.1) means that numerous host factors are potential drug targets. Activation of key host cell signalling pathways is dependent on post-translational phosphorylation events mediated by kinases, and thus kinase inhibitors have been largely implicated in IE gene expression inhibition. Host “epigenetic” factors are also potential targets to lock down IE transcription and hence inhibit viral replication and reactivation from latency. In addition, viral proteins are known to be directly involved in IE gene expression or regulation of host cell signalling pathways exploited by HCMV to activate IE gene expression. However, compounds targeting these viral proteins are considered beyond the scope of this review, as they would contribute to their own specific drug classes. 

Small molecules that inhibit IE gene expression have been identified using a variety of strategic approaches; (i) exploitation of existing compounds that inhibit cell signalling pathways modulated by HCMV infection to facilitate MIEP activation and IE gene expression, (ii) targeted-screening of compound libraries composed of molecules that inhibit key cellular signalling components, e.g., kinase inhibitors, (iii) cell-based assays designed to discover novel molecules that target the early steps of HCMV replication and (iv) testing of compounds that have anecdotal evidence suggesting that they may have antiviral activity. Groups of related compounds have been identified using a combination of these approaches. Key groups of molecules are discussed below, although it should be noted that, in general, the small molecules that have thus far been identified have not had their mechanism of action fully elucidated. 

### 6.2. Artemisinin and Derivatives

Testing compounds that have anecdotal evidence suggesting that they may have antiviral activity often revolves around natural products. Various natural products have been reported to have anti-HCMV activity linked to inhibition of IE expression or function, but most remain largely unsubstantiated beyond initial observations. However, a considerable body of evidence has been generated with respect to the anti-HCMV activity of natural product artemisinin, its semi-synthetic derivative artesunate and various related compounds. Artemisinin is a natural product derived from the plant *Artemisia annua* (Sweet Wormwood), a herb used in traditional Chinese medicine [317,318]. Artemisinin and its derivatives are best known for effective antimalarial activity and treatment [317,318], which provided the premise for testing artesunate for anti-HCMV activity [319]. Artesunate, along with various related compounds, exhibit in vitro inhibitory activity against laboratory, clinical and drug-resistant strains of HCMV in a range of cell types with EC_50_ values generally in the low micromolar to sub-micromolar range [319,320,321,322,323,324,325,326,327,328,329]. Chemically linking artemisinin-related molecules into dimers and trimers significantly improves antiviral potency [320,324,325,326,330,331]. Examples include artemisinin-derived dimer diphenyl phosphate (838), a potent, selective HCMV inhibitor with irreversible activity [332,333] and trimeric artesunate derivative TF27 [326], which exhibits potent in vitro and in vivo activity in the MCMV model [329]. Hybridization of artemisinin-derivatives with bioactive molecules, such as quinazoline, has produced novel compounds with potent anti-HCMV activity significantly better than parental compounds and GCV [334,335,336,337].

The mechanism of action by which artesunate and the various derivatives generate their anti-HCMV activity has not been fully elucidated, but the general consensus is that artesunate primarily interferes with the NF-κΒ pathway [319,326,331]. The NF-κΒ pathway is stimulated upon HCMV infection and activates the MIEP, driving expression of IE proteins and hence subsequent steps in HCMV lytic replication and pathogenesis [179,338,339]. Indeed, artesunate, along with many derivatives, has been shown to block the IE phase of HCMV replication via a reduction in expression levels of IE2, and to a lesser extent IE1 [319,320,326,328,331,332]. Artesunate, and dimer/trimer derivatives such as TF27, have been shown to interfere with the NF-κΒ pathway, which is proposed to occur via a direct interaction of the compound with NF-κΒ subunit RelA/p65 [319,326,331]. Interaction of artesunate with a host cell factor leads to the expectation that acquisition of drug resistance would be less likely; indeed, attempts to generate artesunate drug-resistant isolates in vitro have thus far been unsuccessful [326,340]. Alternative modes of action, implicating other cell signalling pathways and modulation of cell cycle progression, have also been proposed for artesunate compounds [319,340].

Clinical use of artesunate for management of drug-resistant HCMV infections in stem cell or solid organ transplant recipients is considered feasible due the documented anti-HCMV activity discussed above, favourable results in a rodent animal model study [341] and the long and safe clinical history of artesunate treatment in malaria patients [317]. The first use of artesunate in a clinical setting was a success with artesunate being reported as an effective inhibitor of HCMV replication in the treated patient [342]. However, subsequent studies reported either mixed success or that artesunate was ineffective in controlling HCMV infection [343,344,345]. Further studies are required to fully determine the differences in clinical outcomes for artesunate-treated patients, and studies with more potent artesunate derivatives may hold future promise. For example, the trimeric derivative TF27 has recently been demonstrated to display antiviral efficacy in the mouse model. MCMV replication was significantly reduced and restricted to the site of infection, preventing organ dissemination without adverse effects [329].

### 6.3. NF-κΒ Inhibitors

HCMV infection modulates several cell signalling pathways, including the NF-κΒ and PI3K/AKT pathways, in order to facilitate MIEP activation and IE gene expression (see Section 2.1) [183]. Artemsinin and derivatives, which interfere with the NF-κΒ pathway, were discussed in Section 6.2. The mode of action of these compounds was identified after testing for anti-HCMV activity based on their anti-malarial properties. An alternative strategy for identification of anti-HCMV compounds that inhibit major IE gene expression, is to exploit existing compounds already known to inhibit cell signalling pathways modulated by HCMV. A rich source of compounds that could be repurposed as anti-HCMV compounds are the numerous NF-κΒ pathway inhibitors that have been identified for reasons unrelated to HCMV [346,347]. For example, IKK2 inhibitor AS602868 targets a crucial step in NF-κΒ pathway activation: the phosphorylation and subsequent degradation of IκΒ by the IKK complex [348,349]. Testing of AS602868 showed that this compound prevents HCMV mediated NF-κΒ pathway activation, resulting in significant inhibition of IE gene expression, HCMV replication and HCMV-induced host cell inflammatory response without cytotoxicity [350]. HCMV infection also up-regulates the PI3K/AKT pathway leading to activation of NF-κΒ in a PI3K-dependent manner. LY294002, a PI3K inhibitor, significantly reduces HCMV IE1/2 expression, viral DNA replication and viral titers [156,351]. Disruption of the PI3K pathway and subsequent AKT and NF-κΒ activation has been suggested as a possible mechanism of action for heat shock protein 90 (hsp90) inhibitors geldanamycin and 17AAG, which significantly inhibit HCMV replication by affecting IE protein production and hence subsequent steps in HCMV productive replication [351,352]. The examples discussed above demonstrate the value of repurposing existing cell signalling pathway inhibitors for targeting HCMV by inhibition of major IE gene expression. 

### 6.4. Kinase Inhibitors

A general theme in host cell signalling pathway inhibitors is to target kinases, which mediate regulatory post-translational phosphorylation modifications of pathway components. There is an abundance of kinase inhibitors, which have been identified and developed for a wide range of applications particularly cancer treatment [353], which can potentially be repurposed as anti-HCMV compounds. Indeed, examples of kinase inhibitors (AS602868, LY294002) with activity against HCMV have already been discussed in Section 6.3. A further example of a kinase inhibitor repurposed for anti-HCMV testing is the multi-targeted anti-cancer tyrosine kinase inhibitor sorafenib (Nexavar), which has been shown to inhibit MIEP activity, IE expression and also later stages of HCMV replication [172]. The mechanism by which sorafenib inhibits HCMV was not fully elucidated due to its multitude of known kinase targets. However, inhibition of RAF1 activation was implicated but via a mechanism independent of MAPK/ERK signalling [172]. Inhibitors of CDKs also have potential as antiviral drug candidates; for example, the CDK7 inhibitor LDC4297 blocked HCMV replication with EC_50_ values in the nanomolar range [175]. The compound’s mode of action was concluded to be multifaceted but occurs at the level of IE gene expression and interferes with HCMV-mediated inactivation of the retinoblastoma (Rb) protein, which controls progression through the G1 phase of the cell cycle via its phosphorylation state and ability to bind transcription factor complexes [175]. Promisingly, LDC4297 has been shown to possess in vivo antiviral activity in the mouse model. MCMV replication was significantly reduced and restricted to the site of infection, preventing organ dissemination without adverse effects [329].

In addition to directly repurposing known kinase inhibitors, a number of cell-based screens have been performed against various targeted kinase inhibitor compound libraries [354,355,356,357]. Compound-treated HCMV-infected cells were monitored for antiviral effects via expression of a green fluorescent protein (GFP) reporter [354] or late viral protein pp28 from the HCMV genome [355,356,357]. These screens have identified a number of interesting kinase inhibitors with anti-HCMV activity against laboratory and clinical strains that target a variety of cellular kinases without causing significant cytotoxicity. The lack of kinase inhibitor target specificity has made full elucidation of mechanism of action challenging, although in all cases discussed here antiviral activity has been linked to interference with IE expression or protein production without affecting viral entry. Several c-Jun N-terminal kinase (JNK) inhibitors were identified following a 600 compound kinase inhibitor library screen and the SP600125 inhibitor was shown to inhibit JNK activation and suppress IE gene transcription [354]. XMD7 5-aminopyrazine compounds were identified upon screening of the Gray kinase inhibitor library. These compounds target a range of cellular protein kinases to inhibit HCMV replication via a reduction in genome-wide transcription and a defect in the production of certain HCMV proteins including IE2 (86kDa, 60kDa and 40kDa species) [355]. The proposed mechanism of action for XMD7 compounds is consistent with IE2′s role as a viral transcriptional activator, but it is not clear why only a subset of HCMV proteins is affected. CMGC kinase inhibitor RO0504985, an oxindole compound with anti-HCMV activity identified by screening a Roche kinase inhibitor library, also inhibited IE2 and pp28 protein levels [356]. Screening of the GlaxoSmithKline kinase inhibitor set identified SB-734117, a furazan benzimidazole compound, which inhibits several proteins from the AGC and CMCG kinase groups [357]. SB-734117 inhibited IE protein production and reduced phosphorylation of host cell transcription factor CREB and histone H3. However, disappointingly these effects did not lead to any defects in transcription from the MIEP and thus the compound’s mechanism of action remains undetermined [357]. Overall, the wealth of preexisting kinase inhibitors and accompanying knowledge offers good potential for the identification and development of novel anti-HCMV compounds. 

### 6.5. Histone Modifying Enzyme Inhibitors

Major IE gene expression is also regulated by “epigenetic” modifications, including histone post-translational methylation, which can result in repressed gene expression [358]. Histone demethylases are required to remove repressive “epigenetic” marks to promote IE gene expression and hence HCMV productive infection or reactivation from latency [358]. Histone demethylase inhibitors (e.g., ML324, a JMJD2 demethylase family inhibitor) have been shown to potently inhibit HCMV IE gene expression [123,359,360]. These demethylase inhibitors also repress IE gene expression from the related Herpes simplex virus type 1 (HSV-1), and importantly they have been shown to potently inhibit HSV-1 infection and reactivation from latency [123,359,360]. These results suggest that compounds that target histone demethylases and possibly other histone modifying or chromatin remodeling enzymes may have potential as HCMV inhibitors [358].

### 6.6. Cardiac Glycosides

Discovery of novel small molecules that inhibit major IE gene expression can be accomplished using cell-based assays designed to target the early steps of HCMV replication including IE expression but also virus attachment, entry and capsid transport steps [361,362]. One such assay utilizes an engineered variant of the HCMV laboratory strain AD169 that expresses IE2 with a C-terminal yellow fluorescent protein tag (AD169_IE2-YFP_) [362]. IE2-YFP levels in the nucleus of infected cells are quantified using high-content confocal microscopy and hit inhibitory compounds identified via a decrease in nuclear fluorescent signal and therefore a decrease in IE2-YFP protein levels. This IE2-YFP cell-based reporter assay was used to screen a 2080 bioactive compound library and identified one lead compound, the cardiac glycoside convallatoxin. This compound exhibited potent anti-HCMV activity (EC_50_ values in the low nanomolar range) without significant cellular cytotoxicity [362,363]. However, it should be noted that convallatoxin has been discounted as a hit from a different screen due to toxicity [364]. Interestingly, other cardiac glycosides (ouabain, β-antiarin, digoxin, digitoxin) have also been reported to exhibit anti-HCMV activity [363,365,366,367]. Inhibition of HCMV by cardiac glycosides is effective against clinical and GCV-resistant strains and exhibits additive activity when administered to cells in combination with GCV [362,363,365,366]. Members of this compound family have been used clinically for the treatment of heart conditions such as congestive heart failure, although toxicity and dosage issues mean that they are increasingly replaced with synthetic drugs such as ACE inhibitors and beta-blockers [368]. Clinical development as antiviral drugs has not yet been undertaken, but medicinal chemistry approaches have demonstrated the ability to improve antiviral activity and selectivity [369].

A common feature of cardiac glycoside treatment is reduction in IE1/2 protein levels [362,363,365]. Mechanism of action studies demonstrated that convallatoxin does not inhibit IE2 mRNA levels but instead inhibits global translation of viral and host cell proteins [363]. Cellular translation machinery is not directly inhibited by convallatoxin; instead the compound reduces methionine transport into the cell, limiting the intracellular pool of this essential amino acid for translation. Convallatoxin has been proposed to mediate this indirect mechanism of translation inhibition by its ability to bind to and inhibit the cellular sodium-potassium ATP pump (NA^+^,K^+^-ATPase) [363]. In this model, inhibition of the pump causes a reduction in the sodium gradient across the cell membrane, leading to a decrease in sodium-dependent methionine transport [363]. Despite inhibition of global translation, minimal cellular cytotoxicity was observed at the nanomolar concentrations of convallatoxin required for antiviral effect. This observation suggests that, whilst the cell can tolerate a reduction in protein synthesis, HCMV is unable to compensate for reductions in viral protein levels, particularly in IE proteins which are required for early and late protein production and are thus essential for virus replication [363]. Convallatoxin-induced inhibition of viral protein translation by methionine transport reduction is not the only mechanism attributed to the antiviral activity of cardiac glycosides. Alternate mechanisms of action are based on the ability of these compounds to modulate cell signalling pathways [370]. For example, cardiac glycoside digitoxin has been reported to inhibit HCMV through induction of cellular autophagy following activation of the regulatory kinase AMPK via a novel NA^+^,K^+^-ATPase subunit α1-AMPK-ULK1 pathway [173]. In addition to inhibiting HCMV, cardiac glycosides act as antivirals against a range of clinically important DNA and RNA viruses. This broad-spectrum activity has been attributed to a range of host-directed mechanisms [371]. 

### 6.7. Novel Miscellaneous Compounds

A cell-based screen designed to monitor IE2 nuclear translocation was used to identify the cardiac glycoside convallatoxin discussed in Section 6.6. A second screening approach targeting IE2 gene expression utilized a reporter cell-line in which the IE2-activated HCMV TRL4 promoter drives luciferase expression [361]. The reporter cell-line assay was used to screen a 9600 compound library for inhibitors of early phase HCMV infection [361]. Two hit compounds arising from the screen, 1-(3,5-dichloro-4-pyridyl)piperidine-4-carboxamide and 2,4-diamino-6-(4-methoxyphenyl)pyrimidine termed DPCC and 35C10, respectively, have been demonstrated to potently inhibit HCMV replication as effectively as GCV [361,372]. DPCC was also independently identified as having potent anti-HCMV activity in an unrelated high throughput screen designed to target IE1 IFN antagonist function (assay concept described in Section 7.2) [373]. In this screen, DPPC was alternatively termed StA-IE1-3, and a further novel hit compound with similar anti-HCMV activity termed StA-IE1-2 (1-(3-nitrophenyl)-2-(pyrido[3,2-d][1,3]thiazol-2-ylthio)ethan-1-one) was also identified [373]. All three structurally diverse compounds act after viral entry but before IE expression, with significantly decreased IE1 and IE2 expression at both the mRNA and protein levels [361,372,373]. Like the majority of compounds that have been shown to inhibit HCMV IE gene expression, the precise mechanism of action of these three compounds has not been elucidated. However, it has been postulated that they may target a cellular transcription factor or upstream signalling protein required for activation of the HCMV MIEP [373].

### 6.8. Summary

Overall, a variety of approaches utilizing existing knowledge to repurpose known compounds or screening campaigns to discover novel compounds has successfully identified a wide variety of IE gene expression inhibitors. These inhibitors exploit host cell factors and signalling pathways utilized by HCMV to activate the MIEP and thus offer the hypothetical advantage of a reduced risk of drug resistance. A major challenge associated with the development of these compounds is the complexity in elucidating their relevant host cell targets and mechanism of action. Characterization of these compounds has been predominantly conducted in vitro. However, a few compounds have been progressed to in vivo testing using the MCMV model, and clinical testing of artesunate produced mixed clinical outcomes that warrant further investigation. Overall, compounds that inhibit HCMV IE gene expression merit future investigation and development as potential antivirals.

## 7. Inhibition of Major IE Protein Functions

### 7.1. IE2 Inhibitors

Compounds that inhibit IE2 function have been identified and offer promise as a potential new class of HCMV inhibitors (Figure 2). IE2 has been targeted, as it is an essential multifunctional protein that regulates critical events in HCMV replication including transactivation of early and late genes and auto-regulation of the MIEP. IE2 has also been linked to broad dysregulation of host gene expression affecting cell cycle progression, immunomodulation and pathogenesis (see Section 3.1). The first compound demonstrated to directly inhibit IE2 function was WC5, a 6-aminoquinolone derivative [374]. WC5 was tested based on evidence that compounds within this chemical group exhibit antiviral activity against HIV-1 by inhibiting Tat transactivation [375,376]. WC5 specifically inhibits HCMV but not a selection of other herpesviruses [376,377]. In addition to IE, early and late gene expression profiles, which suggest inhibition of IE2 function, WC5 has been shown to directly inhibit IE2′s transactivating activity via a cell-based assay in which an EGFP reporter gene was placed under the control of IE2-dependent early gene promoters [374]. In these assays, WC5 significantly inhibited IE2-mediated transcriptional activation of early gene promoters UL54 and UL112/113. A minimal region of the UL54 promoter composed of a 150-bp segment upstream of the transcriptional start site has been demonstrated to be sufficient to mediate the inhibitory activity of WC5 [378]. Within this 150-bp segment is the IR-1 signal (8-bp inverted repeat element 1), a *cis*-acting sequence with an established role in IE2-dependent transactivation, yet the IR-1 signal has been shown not to be required for WC5′s inhibitory activity [378]. In addition, two key protein interactions, IE2 dimerization and its interaction with TBP, known to be involved in IE2-dependent transactivation of viral promoters have also been discounted as WC5′s target [378]. Intriguingly, WC5′s activity appears to be specifically confined to the regulation of HCMV promoters, as the compound exhibits no effect on a variety of cellular promoters regulated either by IE2 protein interactions or a direct IE2 interaction with promoter DNA [378]. In addition to inhibiting IE2 transactivation of viral promoters, a second mechanism of action by which WC5 inhibits a different IE2 function has been identified [378]. WC5 specifically disrupts IE2′s direct interaction with the crs within the MIEP (Figure 1). Disruption of the IE2-crs interaction abolishes IE2′s auto-repression of its own promoter, a function essential for viral replication. Although WC5 has been demonstrated to inhibit two IE2 functions, transactivation of viral early and late genes and MIEP auto-regulation, the exact molecular mechanisms require further elucidation.

WC5′s unique activity offers the possibility to develop a new mechanistic class of anti-HCMV compounds. Towards this goal, WC5 potently and selectively inhibits HCMV replication in the sub-micromolar range irrespective of testing against laboratory or clinical isolates, and its activity is comparable to GCV [376]. Unsurprisingly, given WC5′s novel mechanisms of action, the compound similarly inhibits isolates resistant to clinically approved anti-herpesvirus DNA polymerase inhibitors [376]. Further, when WC5 is combined with GCV, synergistic activity against HCMV replication was observed without significant increases in cellular cytotoxicity [374]. WC5 also inhibits MCMV replication, albeit with ~10-fold lower activity compared to HCMV [376]. Importantly, WC5′s mechanism of action against HCMV and MCMV appears to be conserved, as it has been shown to block MCMV early gene transactivation mediated by the MCMV IE2 homolog ie3 [378]. Thus, it has been suggested that the murine model may be used to test WC5 activity in vivo as a prerequisite to clinical development [378]. Structure–activity relationship studies have been conducted with the aim of improving WC5′s potency [374,379]. These studies gained insight into chemical groups required for WC5 activity, and identified an analogue with an improved selectivity index compared to WC5 without compromising antiviral activity. However, analogues with significantly improved potency were not identified [379].

WC5′s discovery together with its novel mechanism of action provided proof-of-principle that IE2 is a valid target for drug discovery and encouraged a screening campaign to identify new compounds targeting IE2 [364,380]. A screen has been performed employing essentially the same cell-based assay used to determine WC5′s mechanism of action as an inhibitor of IE2-mediated transactivation of early gene expression [374]. Assay optimization identified conditions using the stable cell-line that expresses EGFP under the control of the IE2-dependent UL54 early promoter as suitable for screening purposes [364]. A 2320 bioactive compound library including all FDA-approved drugs was screened and six hit compounds have so far been selected for further study [364,381,382]. These hit compounds are deguelin (DGN), nitazoxanide (NTZ), thioguanosine (TGN), alexidine dihydrochloride (AXN), manidipine dihydrochloride (MND) and berberine (BBR). All hits inhibited HCMV replication with EC_50_ values in the low micromolar range and lacked significant toxicity. This antiviral activity was observed for laboratory, clinical and drug-resistant HCMV isolates. Further, MND was shown to be inactive against a selection of other DNA and RNA viruses and is thus likely to be a specific anti-HCMV compound [381]. The antiviral mechanism of these compounds was confirmed to be inhibition of IE2-mediated viral early gene transactivation and, like with WC5, a minimal 150-bp segment of the UL54 promoter is sufficient for inhibitory activity. However, the precise mechanism of action has not been elucidated, although prior knowledge of these bioactive compounds has led to the proposal that they are likely to interfere with pathways in HCMV-infected cells that are required for the switch from the IE to early phase of viral replication [364]. Despite the lack of a precise mechanism of action, repurposing of bioactive compounds for anti-HCMV activities may allow compound development to be fast-tracked, especially in the case of MND, as it is already an FDA-approved drug used in the treatment of hypertension [381].

### 7.2. IE1 Inhibitors

Identified inhibitors of IE2-dependent transactivation do not inhibit IE1-dependent transactivation and are thus specific to IE2 not IE1 function [364,382,383]. IE1′s function as an IFN antagonist has been targeted for drug discovery via a modular cell-based screening platform designed to identify compounds that inhibit a viral IFN antagonist choice [373]. The platform utilizes two reporter cell-lines that provide a simple method to detect activation of IFN induction or signalling via an EGFP gene placed under the control of the IFN-β or an ISRE-containing promoter, respectively. IE1 counteracts type I IFN signalling by binding directly to STAT2 thereby preventing the ISGF3 transcription factor from binding ISRE elements in the promoters of ISGs (see Section 3.4) [225]. Therefore, a derivative of the IFN signalling reporter cell-line expressing IE1 was generated that blocks EGFP expression upon IFN signalling pathway activation. This reporter cell-line was used to screen a 16,000 compound library to identify compounds that release the IE1-imposed IFN response block and hence restore EGFP expression [373]. Two hit compounds, StA-IE1-2 and StA-IE1-3, were identified and demonstrated to be potent inhibitors of HCMV replication [373]. Compound characterization revealed that, instead of identifying anticipated IE1 IFN antagonist function inhibitors that target at the protein level, the compounds act at the mRNA level and interfere with IE1/2 transcription. The likely explanation is that IE1 expression was driven by MIEP sequences in the lentiviral vector used to create the IE1 reporter cell-line derivative. StA-IE1-2 and -3 are therefore also described in Section 6.7 concerning inhibition of IE gene expression. Despite the unexpected results, the assay platform had previously been used to identify compounds that specifically target the IFN antagonist function of Respiratory Syncytial Virus non-structural protein 2 [373]. To the best of our knowledge, no other IE1-specific drug discovery screens have been undertaken.

### 7.3. Summary

Overall, strategies to target IE2 function have identified a number of interesting compounds, although their exact mechanism of action has not been fully elucidated. To date, compounds that target IE1 function have not been identified, and IE1 remains an important but underexploited potential drug target.

## 8. Conclusions and Future Perspectives

This review provides an update on the regulation of HCMV major IE gene expression and IE1 and IE2 protein functions. We discussed existing clinically approved therapies and why major IE gene expression and IE1/2 protein functions are considered potential alternative targets for anti-HCMV strategies. We outlined the various molecular and chemical approaches that are being used to target major IE gene expression and protein function. Advances in molecular approaches, particularly genome-editing technology, are opening up new promising strategies for targeting HCMV. However, further research and development are required before this novel technology can be translated clinically. Key groups of small chemical inhibitors targeting major IE gene expression or IE1/2 protein function are highlighted in the review. A major challenge associated with the vast majority of these compounds is the complexity in elucidating their relevant targets, many of which are host cell proteins, and mechanism of action. Basic research to determine this knowledge will highlight the value of these compounds as chemical tools to further understand regulation of major IE gene expression and/or IE1/2 protein functions. It will also promote further in vivo and clinical testing of these molecules, which is currently limited to only a few studies and compounds. A key advantage of targeting major IE gene expression and IE1/2 function is the potential to inhibit reactivation from latency, a property that existing therapies, which target viral replication, do not achieve. However, testing key compounds for this attribute is mostly lacking due to the specialized methodology and limitations of in vitro cell-based latency models. Yet, the identification of compounds that repress latency is becoming more pressing as organ and haematopoietic stem cell transplantation has become more common. Existing anti-HCMV drugs are also not approved for use during pregnancy because of their teratogenic and embryotoxic effects in animal studies, yet the need for antiviral therapy in congenitally infected neonates for improved long-term outcomes is increasingly appreciated. Finally, IE1/2-directed drugs are not expected to confer cross-resistance to or interfere with the activity of existing drugs currently approved for HCMV monotherapy. They may therefore be combined with these drugs for combination therapies with improved efficacy. Overall, “bright and early” events in HCMV infection deserve more attention as a promising antiviral strategy against HCMV.

## Figures and Tables

**Figure 1 viruses-12-00110-f001:**
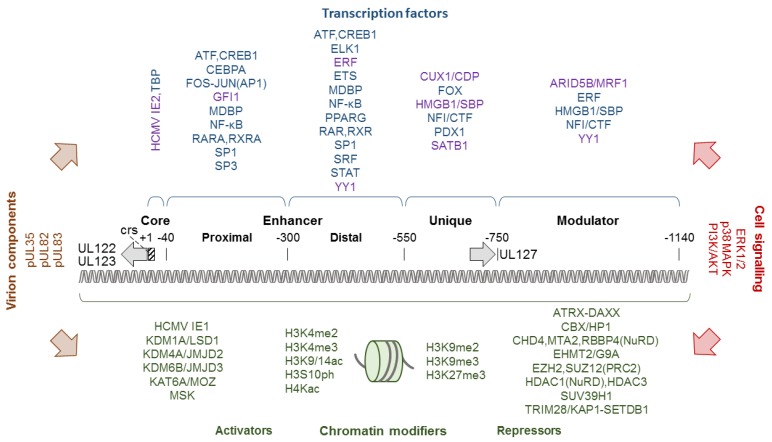
Organisation of the human cytomegalovirus (HCMV) major IE enhancer and promoter (MIEP) and select protein factors involved in its regulation. The MIEP is composed of a core promoter containing a TATA-box and the crs that mediates repression by IE2, an enhancer with proximal and distal parts, a unique element and a modulator. Nucleotide positions relative to the transcription start sites and the direction of transcription (grey arrows) are indicated. “Leftward” transcription results in mRNAs encoding the IE1 and IE2 proteins (“rightward” transcription results in uncharacterized mRNAs containing the UL127 open reading frame). Transcription factors known or predicted to bind to the individual parts of the MIEP are shown above (repressors are shown in purple). Chromatin modifiers and histone tail modifications reported to activate or repress the MIEP are shown below. A few examples of virion components and cell signalling pathways known to activate the MIEP are shown at the left and right side, respectively, of the diagram. ARID5B/MRF1, AT-rich interaction domain 5B protein; ATF, activating transcription factor family; CBX/HP1, heterochromatin protein 1; CEBPA, CCAAT enhancer binding protein alpha; CHD4, chromodomain helicase DNA binding protein 4, nucleosome remodeling and deacetylase (NuRD) subunit; CUX1/CDP, cut-like homeobox 1 protein; ELK1, ETS transcription factor Elk1; ETS, Ets proto-oncogene transcription factor; EZH2, enhancer of zeste 2 polycomb repressive complex 2 (PRC2) subunit; FOS, Fos proto-oncogene, activator protein 1 (AP-1) transcription factor subunit; FOX, forkhead transcription factor family; GFI1, growth factor-independent 1 transcriptional repressor; HMGB1/SBP, high mobility group box 1 protein; JUN, Jun proto-oncogene, AP-1 transcription factor subunit; KAT6A/MOZ, lysine acetyltransferase 6A; KDM1A/LSD1, lysine demethylase 1A; KDM4A/JMJD2, lysine demethylase 4A; KDM6B/JMJD3, lysine demethylase 6B; MDBP, methylated DNA binding protein family; MTA2, metastasis-associated 1 family member 2, NuRD subunit; NFI/CTF, nuclear factor 1 family; PDX1, pancreatic and duodenal homeobox 1 protein; PPARG, peroxisome proliferator-activated receptor gamma; RARA, retinoic acid receptor alpha; RBBP4, Rb binding protein 4 chromatin remodelling factor, NuRD subunit; RXRA, retinoic X receptor alpha; SATB1, special AT-rich sequence binding homeobox 1 protein; SETDB1, SET domain bifurcated histone lysine methyltransferase 1; SP1, Sp1 transcription factor; SP3, Sp3 transcription factor; SRF, serum response factor; SUZ12, suppressor of zeste 12 PRC2 subunit; TBP, TATA-box binding protein; TRIM28/KAP1, tripartite motif containing 28 protein. See main text for other abbreviations.

**Figure 2 viruses-12-00110-f002:**
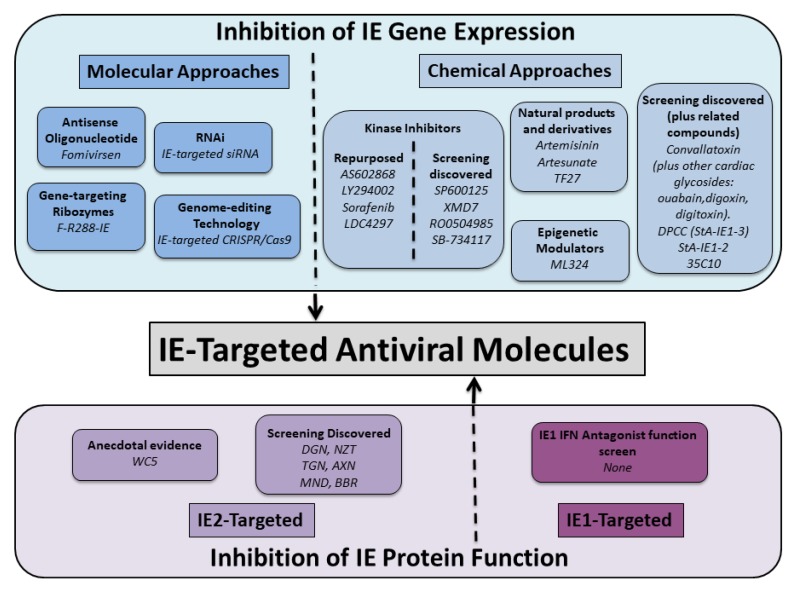
Schematic of molecular and chemical approaches used to target major IE gene expression and IE protein function. Key groups of molecules are listed by category, and examples of molecules within each category given in italics. DGN, deguelin; NZT, nitazoxanide; TGN, thioguanosine; AXN, alexidine dihydrochloride; MND, manidipine dihydrochloride.

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
