# Peer review of "Bright and Early: Inhibiting Human Cytomegalovirus by Targeting Major Immediate-Early Gene Expression or Protein Function"

_viruses, 2020, doi:10.3390/v12010110_

Round 1

Reviewer 1 Report

This is a very comprehensive review article that describes the current state of HCMV antivirals and makes a case for targeting the IE1 and IE2 proteins, in what the authors call a 'Bright and Early' approach.  

Overall, this manuscript is very well written, with strong attention to detail. The figures are clear and easy to interpret. It should serve as a well-cited reference in the field for years to come. 

Minor comments: 

page 2: "The virus-host balance is not yet established or has been disrupted in individuals with immature or compromised immune systems, respectively" - I suggest revising this statement for improved clarity. The paragraph opens with a statement about virus-host co-evolution and how that enables persistent HCMV infection for the life of the host. In this light, the sentence quoted above seems to continue the discussion of evolution, although immunosuppression in the minority of hosts is unlikely to drive viral evolution in the global population. Perhaps it would be clearer just to state that immune system defects disrupt the delicate balance between virus and host and can result in severe disease outcomes. 

The second paragraph on page 2 closes with a statement about the promise of adoptive T-cell transfer for antiviral therapy, which seems out of place in the context of the overall paragraph. Consider revising or striking this statement. 

page 4: "In contrast..." should instead be "By contrast...." 

page 7: More information about the post-transcriptional role of p97 would be welcome here. The current paragraph is a little vague on the details.

page 17: The first line has "...known kinases inhibitors,", which should instead be "kinase".  

Author Response

We thank the reviewer for carefully reading the manuscript and for his/her positive comments.

The sentence "The virus-host balance is not yet established or has been disrupted in individuals with immature or compromised immune systems, respectively" on page 2 has been changed to "Developmental or acquired immune system defects disrupt the delicate balance between virus and host and can result in severe disease outcomes".

The statement on adoptive T-cell transfer on page 2 has been revised ("Since cell-mediated adaptive immunity is believed to be key in counteracting HCMV infection, adoptive transfer of virus-specific T cells holds promise for antiviral therapy [57-59]") and moved to the subsequent paragraph.

On page 4, "In contrast" has been changed to "By contrast".

A sentence on the role of p97 in post-transcriptional regulation of the major IE gene has been added to page 7: "RNA sequencing showed increased IE1 and decreased IE2 splicing following p97 knockdown [187]".

"Known kinases inhibitors" has been changed to "known kinase inhibitors" on page 17.

Reviewer 2 Report

This is a very well written review with excellent references and clear outline of the background and ultimate goal for treating CMV.

I only picked up a few places for correction which include:

On Page 8, the first line of section 3.2 reads "While IE2 is known to bind sequence-specifically to DNA...." this needs to be reworded as it makes no sense to me.

On page 19, section 7.1 has the sentence "In addition to IE, early and late gene expression profiles suggesting inhibition of IE2 function....." do the authors mean to say "suggest" inhibition of IE2 function or did they leave something out here?

On bottom of page 19 last sentence I think you need a comma after GCV

On page 20 section IE1 inhibitors: In the first line it says " Identified inhibitors of IE2-dependent transactivation do not to inhibit IE1..."

I think you need to remove the word "to"

Otherwise good to go!

Author Response

We thank the reviewer for carefully reading our manuscript and for the positive feedback he/she has provided.

The sentence on page 9 "While IE2 is known to bind sequence-specifically to DNA, there is no convincing evidence for direct DNA binding by IE1" was reworded to "IE2 is known to bind sequence-specifically to DNA, but there is no convincing evidence for direct DNA binding by IE1".

On page 19, the sentence "In addition to IE, early and late gene expression profiles suggesting inhibition of IE2 function, ..." was changed to "In addition to IE, early and late gene expression profiles, which suggest inhibition of IE2 function, ...".

A comma was inserted after "Further, when WC5 is combined with GCV..." on page 20 (the only "GCV" on page 19 is at the end of a sentence).

On page 20, "to" was removed from the sentence "Identified inhibitors of IE2-dependent transactivation do not to inhibit IE1...".